# Daily high-frequency transcranial random noise stimulation (hf-tRNS) for sleep disturbances and cognitive dysfunction in patients with mild vascular cognitive impairments: A study protocol for a pilot randomized controlled trial

**Yuqi Gong**[1], **Jing Li**[1], **Yuk Shan Yuen**[1,2], **Natalie Shu Yang**[3], **Zeyan Li**[1], **Wai Kwong Tang**[1], **Hanna Lu**[1,4]*

**1** Department of Psychiatry, The Chinese University of Hong Kong, Hong Kong SAR, China, **2** Department of Linguistics and Modern Languages, The Chinese University of Hong Kong, Hong Kong SAR, China, **3** Department of Psychology, University of Greifswald, Greifswald, Germany, **4** The Affiliated Brain Hospital of Guangzhou Medical University, Guangzhou, China

* hannalu@cuhk.edu.hk

**Data Availability Statement:** Data cannot be shared publicly because of personal medical

# Abstract

## Background

Poor sleep quality is increasingly considered to be an underlying cause of cerebrovascular diseases. This is a slowly progressing condition that gradually leads to vascular cognitive impairment and stroke during ageing. At present, randomized clinical trials examining the non-pharmacological therapies in the management of this comorbidity are very limited. Transcranial current stimulation (tCS) is a non-invasive technology for promoting cognitive function and treating brain disorders. As advanced modalities of tCS, transcranial random noise stimulation (tRNS) and transcranial alternating current stimulation (tACS), could deliver frequency-specific waveforms of currents that can modulate brain activities in a more specific manner.

## Methods and design

Chinese individuals between the ages of 60 and 90 years, who are right-handed and have mild vascular cognitive impairment (VCI) with sleep disturbances, will participate in a randomized study. They will undergo a 2-week intervention period where they will be randomly assigned to one of three groups: high-frequency (hf)-tRNS, 40 Hz tACS, or sham tCS. Each group will consist of 15 participants. Before the intervention, high-resolution magnetic resonance imaging (MRI) data will be used to create a computational head model for each participant. This will help identify the treatment target of left inferior parietal lobe (IPL). Throughout the study, comprehensive assessments will be conducted at multiple time points, including baseline, 2nd week, 6th week, and 12th week. These assessments will evaluate various

information. Data are available from the CUHK Data Access / Ethics Committee (contact via hannalu@cuhk.edu.hk) for researchers who meet the criteria for access to confidential data.

**Funding:** This clinical trial is supported by the Direct Grant for Research, The Chinese University of Hong Kong (Grant No. 2024.047). HL received the award. The funders had no role in study design, data collection and analysis, decision to publish, or preparation of the manuscript.

**Competing interests:** The authors have declared that no competing interests exist.

factors such as sleep quality, domain-specific cognitive performance, and actigraphic records. In addition, the participants' adherence to the program and any potential adverse effects will be closely monitored throughout the duration of the intervention.

## Conclusions

The primary objective of this study is to examine the safety, feasibility, and effectiveness of hf-tRNS and 40 Hz tACS interventions targeting left IPL in individuals with mild vascular cognitive impairment (VCI) who experience sleep disturbances and cognitive dysfunction. Additionally, the study seeks to evaluate the program's adherence, tolerability, and any potential adverse effects associated with frequency-specific transcranial current stimulation (tCS). The findings from this research will contribute to a deeper understanding of the intricate relationship between oscillation, sleep, and cognition. Furthermore, the results will provide valuable insights to guide future investigations in the field of sleep medicine and neurodegenerative diseases.

## Trial registration

ClinicalTrials.gov Identifier: NCT06169254.

## Introduction

Sleep is a fundamental biological requirement for brain health and fitness. Poor sleep quality, as a modifiable risk factor, can severely jeopardize the cognitive functions and healthy longevity in ageing populations [1,2]. Of note, the status of sleep disturbances is increasingly considered to be an underlying cause of cerebrovascular diseases, such as stroke and vascular dementia [3]. This is a slowly progressing preclinical condition that gradually leads to impaired cognition and reduced quality of life with ageing.

In the 1960s, Dr. Paul Nogier from GLEM (Lyon) provided a description and practical demonstration showing the significance of oscillations and corrections for overall well-being. Growing evidence indicates that the disruption of sleep slow oscillations (SSOs) can interfere with the strengths of specific synapses tagged as relevant neural circuits, adding to cognitive dysfunction in the individuals with high risk of vascular dementia and stroke [4]. Recent research suggests that slow oscillation activities can facilitate neurons in the process of cleaning toxic materials, removing them through intracranial cerebrospinal fluid (CSF) transport and glymphatic system [4,5], enhancing the sleep quality and cognitive functions correspondingly. This bidirectional relationship between slow oscillations, sleep quality and cognitive changes emphasizes the potential to reverse this phenomenon by modulating brain activity [6]. In addition, sleep, especially slow-wave sleep, plays a pivotal role in modulating cytokine levels and inflammatory response. Sleep deprivation has been shown to activate inflammatory pathways and trigger the release of cytokines, (IL-6), IL-1 beta (IL-1β), and tumor necrosis factor alpha (TNFα), in blood samples has been associated with poor sleep quality [7].

Blood sample testing is typically invasive; however, saliva has emerged as a promising alternative for biomarker detection. Recent studies have unveiled a noteworthy correlation between IL-6, IL-1β and TNF-alpha in both biofluids [8,9]. Despite this, the relationship between cytokine levels in saliva and blood remains a subject of variance. While some studies found no significant links between TNF-α and IL-1β levels in saliva and blood, it was IL-6 that notably

demonstrated a strong correlation across both mediums [10]. In a separate study, chronic renal failure patients with higher PSQI levels exhibited elevated TNF and IL-6 levels compared to the control group [11]. Notably, other study reported IL-1β levels in saliva consistently mirrored serum levels [12]. Additionally, while salivary IL-6 levels showed no significant association with sleep quality, IL-1β was notably linked to PSQI scores. Intriguingly, salivary TNF-alpha inversely correlated with self-perceived health status [13].

Despite the intricate nature of these findings, chronic inflammatory responses have been implicated in a range of health conditions, including cardiovascular disease, metabolic disorders, neurodegenerative diseases, mood disorders, and impaired cognitive function [3,14,15]. Given the complexity of these associations, it is imperative to explore the role of chronic inflammation in sleep disturbances further. Continued research in this domain is crucial to unravel the intricate interplay between sleep quality and salivary cytokines.

Currently, non-pharmacological interventions are recommended as the initial approach for managing sleep disturbances according to clinical guidelines [16]. However, there is a scarcity of clinical trials investigating non-pharmacological therapies, particularly in individuals who are at a higher risk of developing stroke and vascular dementia. As a result, there is a lack of sufficient evidence regarding the effectiveness of these therapies in addressing sleep disturbances in this specific population. As novel non-invasive technologies, transcranial alternating current stimulation (tACS) and transcranial random noise stimulation (tRNS), as the advanced modalities of tCS, could deliver the oscillation-specific waveforms of current that could modulate and monitor brain activities in a more specific manner.

Our team has completed a randomized clinical trial to compare the efficacy of 40Hz tACS on subjective sleep quality and cognitive functions in senior adults with preclinical Alzheimer's disease (AD) (ClinicalTrials.gov Identifier: NCT05544201) [17]. In the preliminary results, we used the Pittsburgh Sleep Quality Index (PSQI) to evaluate subjective sleep quality and found that repeated tACS has a significant positive effect on sleep quality compared to sham tCS (PSQI score change: tACS group vs sham tCS: 6.01 vs 3.72, $p<0.001$).

Although promising results of tACS have been observed in preclinical AD patients, the potential biological mechanisms of oscillation-specific tCS on circadian rhythms and glymphatic system, as well as whether oscillation-specific tCS could be employed as a transdiagnostic non-pharmacological treatment in neurodegenerative diseases, are still not known. Moreover, compared to the stimulation with fixed frequency, high-frequency tRNS can add electrical noise to cortical circuits to enhance neural processing and further induce prolonged physiological and excitability changes [18].

The current body of evidence is insufficient to support the implementation of a large-scale randomized controlled trial (RCT) comparing the effects of oscillation-specific transcranial current stimulation (tCS) on sleep quality and domain-specific cognitive functions, such as executive function and attention. There is also a lack of clinical data that would allow for the estimation of the sample size or the assessment of the efficacy and sustainability of oscillation-specific tCS for a full-scale RCT. Thus, this pilot RCT aims to test the safety and efficacy of oscillation-specific tCS (i.e., tRNS and tACS) for sleep disturbances and cognitive dysfunction in mild vascular cognitive impairment patients. This pilot study will also help to determine the sample size needed for a full-scale RCT. The findings of this study will provide valuable clinical evidence that can inform the effect size and personalized modeling of oscillation-specific tCS for age-related brain diseases. Furthermore, the dynamic changes of sleep quality, cognition and glymphatic system function observed in this pilot RCT will be helpful for in-depth understanding the relationship of "brain oscillations, sleep quality and cognition" and guiding the future studies of clinical neuroscience, brain diseases and sleep medicine.

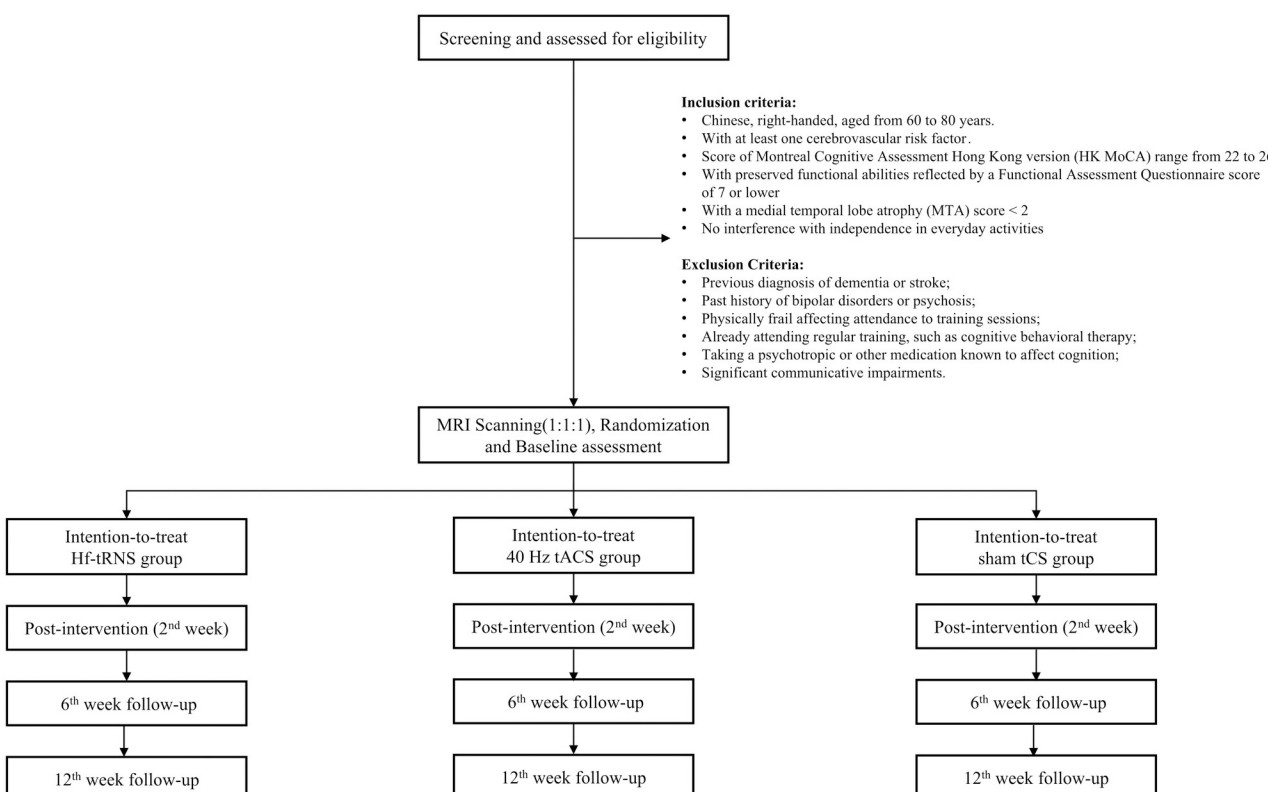

**Fig 1. The Consolidated Standards of Reporting Trials (CONSORT) flow diagram for the clinical trial of oscillation-specific transcranial current stimulation in mild vascular cognitive impairment patients.**

## Materials and methods

### Research design

This clinical trial will be conducted at an academic cognitive training center in Hong Kong. It is a randomized, double-blind study, meaning that both the trial participants and the assessors of the outcomes will be unaware of the assigned conditions. The study will follow a three-arm design with repeated measures taken at baseline, 2nd week, 6th week, and 12th week. The three conditions to be tested are hf-tRNS, tACS, and sham tCS. The study will adhere to the guidelines set forth by the Consolidated Standards of Reporting Trials (CONSORT) statement (http://www.consort-statement.org) [19] (Fig 1) and the Recommendations for Interventional Trials (SPIRIT) [20] (Fig 2). Before participating in the study, written informed consent will be obtained from all participants.

### Sample size and power analysis

The sample size for each study arm is considered to be 15 mild VCI patients, which is appropriate for the primary and secondary study objectives. Aiming for the treatment to be acceptable by 70% of the participants with a precision of 20% (i.e., at least 50% would recommend the treatment), a minimum of 13 participants is required in the treatment arm. To evaluate the potential efficacy of the treatments compared to control group, assuming a medium standardized effect size (0.5), 11 participants are required in each group using an 80% one-sided

| | STUDY PERIOD | | | | |
|---|---|---|---|---|---|
| | Enrolment | Baseline | Post-allocation | | Close-out |
| TIMEPOINT** | -t₁ | 0 | t₁ | t₂ | t₃ |
| **ENROLMENT:** | | | | | |
| Eligibility screen | X | | | | |
| Informed consent | X | | | | |
| Allocation | X | | | | |
| Data collection | | X | X | X | X |
| **INTERVENTIONS:** | | | | | |
| *hf-tRNS* | | •———————• | | | |
| *40 Hz tACS* | | •———————• | | | |
| *Sham-tCS* | | •———————• | | | |
| **ASSESSMENTS:** | | | | | |
| *Sleep quality* | | X | X | X | X |
| *Cognitive function* | | X | X | X | X |

**Fig 2. Schedule of the clinical trial according to the standard protocol items: Recommendations for interventional trials checklist (SPIRIT).** Abbreviations: hf-tRNS = High-frequency transcranial random noise stimulation; tACS = Transcranial alternating current stimulation; tCS = Transcranial current stimulation.

confidence interval approach which is suggested for pilot trials [21,22]. Accounting for a follow-up rate of 10%, the total sample size is calculated as 45.

## Study population, recruitment and eligibility criteria

We will enroll suitable participants from our established research cohort, i.e., the Hong Kong Cohort of Abnormal Sleep in Ageing Population (HK-ASAP) (ClinicalTrials.gov Identifier: NCT06170073). The neuroscientists and psychiatrists in our team will carefully identify individuals who exhibit mild vascular cognitive impairment (VCI). Those who meet the initial criteria will be invited to undergo a screening process led by our trained research assistant. This screening will assess their eligibility and availability to participate in this pilot RCT. Before making a decision and providing informed consent, both the participants and their caregivers will receive an extensive explanation and detailed information about the study.

The Potential mild VCI patients will need to satisfy the following inclusion criteria:

1. Chinese individuals who are right-handed and aged between 60 and 80 years.

2. Participants must have a Montreal Cognitive Assessment Hong Kong version (HK MoCA) score ranging from 22 to 26 [17].

3. Participants should exhibit impaired executive functions as measured by HK MoCA.

4. Participants must have at least one cerebrovascular risk factor, which includes either a history of hypertension (defined as systolic blood pressure of 140 mm Hg or higher, diastolic blood pressure of 90 mm Hg or higher, or receiving antihypertensive medication) or

comorbidity with diabetes mellitus (DM) or hyperlipidemia, or receiving related medication.

5. Based on structural MRI, participants should have a medial temporal lobe atrophy (MTA) score of less than 2, indicating the exclusion of individuals with prodromal Alzheimer's disease.

6. Participants should not experience interference with independence in everyday activities.

The exclusion criteria for participants are as follows:

1. Individuals with a previous diagnosis of dementia or stroke.

2. Participants with a history of bipolar disorders or psychosis.

3. Individuals who are physically frail and may have difficulty attending training sessions.

4. Participants already engaged in regular training, such as cognitive behavioral therapy.

5. Individuals who are using psychotropic medication or other medications known to impact cognition, such as anti-dementia medication.

6. Participants with significant communicative impairments, including severe hearing loss and vision loss.

## Ethical issue

Ethical standards will be strictly adhered to this study, ensuring that informed consent will be obtained from the participants and their anonymity, privacy, and confidentiality are respected. Participants will only be recruited if they are mentally capable of providing consent. Personal identifiers such as names, birth dates, and mobile numbers will not be disclosed in any reports or publications. Participants have the right to withdraw from the study at any time without any interference in their future service use. For those who have any medical concerns during the study, they will be advised to seek help from clinical doctors.

This study has received the initial approval from the Clinical Research Ethics Committee of The Chinese University of Hong Kong (CUHK) and New Territories East Cluster (NTEC) (https://www.crec.cuhk.edu.hk/) on 17th March, 2023 and renewed approval on 22nd February 2024, with the reference number: 2023.074. The study has been registered in the United States National Institute of Health Registration System with Clinical Trials under the registration number NCT06169254 on 13th December 2023. The trial's reporting will adhere to the guidelines set forth by major international journals. Moreover, the study will be conducted in compliance with the principles outlined in the Declaration of Helsinki and the Good Clinical Practice (GCP) guidelines established by the International Conference on Harmonisation (ICH) for the registration of pharmaceuticals for human use (ICH-GCP).

## Neuroimaging

**High-resolution structural MRI.** Neuroimaging scans will take place at the Prince of Wales Hospital utilizing a 3.0 Tesla Siemens MAGNETOM Prisma MRI scanner equipped with a 32-channel head coil. The scans will specifically focus on obtaining high-resolution T1-weighted structural magnetic resonance imaging (MRI) using a Magnetization Prepared RApid Gradient Echo (MPRAGE) sequence. This sequence will be performed with the following parameters: axial acquisition employing a matrix size of 256×256×192, a slice thickness of

1 mm without any gap, a field of view (FOV) measuring 230 mm, a repetition time (TR) of 2070 ms, an echo time (TE) of 3.93 ms, and a flip angle of 15 degrees. By utilizing this sequence, we aim to generate high-quality isotropic images ensuring that each voxel represents a size of 1 mm × 1 mm × 1 mm [23].

**Diffusion tensor imaging (DTI).** To assess the function of glymphatic system, this clinical trial will utilize diffusion tensor imaging (DTI). DTI will involve the application of spin-echo single-shot echo-planar pulse sequences, encompassing a total of 32 different diffusion directions. For DTI, the imaging parameters will include a repetition time (TR) of 11015 ms, an echo time (TE) of 73.5 ms, a 2.5 mm slice thickness, a 128 × 128 acquisition matrix, a field of view (FOV) measuring 224 × 224 mm$^2$, and a b-value of 1000 s/mm$^2$. Additionally, T2 star weighted angiography (SWAN) will be employed with a TR of 43.2 ms, a TE of 4.0 ms, a slice thickness of 2 mm, a 220 × 220 mm$^2$ acquisition matrix, and a flip angle of 20 degrees, as outlined in reference [24].

## Randomization and masking

Participants will be randomly assigned in a 1:1:1 ratio to one of three treatment groups:
1) hf-tRNS, 2) 40 Hz tACS, 3) sham tCS. To ensure equally allocation across different treatment groups, a randomization assignment will be generated prior to enrollment using an online system (http://randomization.com/) by a statistician who is not involved in the study design. Both the assessment staff and participants will be blinded to the study design and group allocation.

## Apparatus and settings

A battery-driven direct current stimulator (DC-Stimulator Plus, NeuroConn, Ilmenau, Germany) will be used to deliver high-definition transcranial current stimulation (HD-tCS). This stimulation will be administered through a central anodal electrode surrounded by four return cathodal electrodes. The base diameter of the HD-tCS electrode will measure 2.4 cm. Consistent with previous studies on cognition and sleep [25,26], the center electrode (anodal) will be positioned over the left inferior parietal lobe (IPL) (i.e., P3 according to the international 10–20 EEG system). The rationale of selecting IPL as the treatment target is because this region is the core part of the default mode network (DMN). Recent research suggests that the suprachiasmatic nucleus (SCN) and default mode network (DMN) work together to regulate the brain's arousal state [27] The SCN is known as the master circadian clock, interacting with various brain regions to regulate sleep-wake cycles. Disruptions in the functional connectivity between the SCN and DMN may contribute to different types of sleep-wake disorders [28]. Focusing on the regions within DMN, medial prefrontal cortex and posterior cingulate cortex are located on the brain's inner surface; IPL is only region of DMN that is located on the outer surface of the brain, making it a suitable treatment target for tCS. Additionally, IPL has been reported to participate in various cognitive processes affected by neurodegenerative diseases, which also makes it a focal point of investigation [29]. To ensure accurate placement of electrodes, individual's structural MRI scans will be utilized to measure and position the electrodes. The electrodes will be securely fixed using conductive paste (Ten20$^®$, Neurodiagnostic Electrode Paste, Weaver and Company, Aurora, CO, USA). Participants will be instructed to relax during the setup of the treatments.

## Stimulation modalities

**High-frequency transcranial random noise stimulation (hf-tRNS).** It will be administered for a duration of 20 minutes at a frequency range of 101–640 Hz and an intensity of 2 milliamps [30].

**40 Hz transcranial alternating current stimulation (tACS).** The stimulation will be applied for 20 minutes at a low frequency between 0.1Hz and 100 Hz, specifically 40 Hz and an intensity of 2 milliamps [30].

**Sham transcranial current stimulation (tCS).** The stimulation duration will be limited to 30 seconds, followed by keeping the electrodes in place for an additional 20 minutes. This procedure is designed to replicate the temporary skin sensation of tingling experienced during hf-tRNS and 40 Hz tACS, while ensuring that no lasting effects are produced [17]. Besides, we also include a skin physiological test that will record the feedback from the participants.

## Grouping and intervention schedule

In this trial, each eligible participant will receive one type of tCS treatment consisting of 10 sessions of oscillation-specific tCS treatment. To ensure a rigorous experimental design, the participants will be randomly allocated to one of the three groups, each representing a different modality of tCS:

1. hf-tRNS (101–640 Hz)

    1. 40 Hz tACS

    2. Sham tCS

This study is designed as a double-blind, sham-controlled, randomized clinical trial. All the participants will be blinded to the group assignment (i.e., treatment). In addition, independent research assistants who are unaware of the group assignments are responsible for gathering data on sleep quality and cognitive functions. The research assistants will also be excluded from outcome assessments to maintain their blinding and ensure unbiased data collection.

## Outcome measures

**Primary outcomes.**

1. *Subjective sleep quality*:

    1. Subjective evaluation of sleep quality
    To assess subjective sleep quality over a one-month period, the Pittsburgh Sleep Quality Index (PSQI) will be utilized. The PSQI is a self-report questionnaire consisting of 19 items [31]. Each item contributes to one of seven component scores, which range from 0 to 3. The maximum total composite score is 21, with higher scores indicating poorer sleep quality. The sum of these component scores provides a measure of global subjective sleep quality. A cutoff score of 5 or more is used to identify poor sleep quality [17]. PSQI has demonstrated adequate reliability in both cognitively intact elderly individuals and patients with dementia [32]. And the validated Chinese version of the PSQI will be employed in this study.

2. *Attentional function*: Complex attention is measured by the Attention Network Test (ANT). The ANT paradigm is implemented using E-Prime 3.0 software [33]. There are four types of cues: no cue, center cue, double cue, and spatial cue. Additionally, there are three types of flankers: neutral, congruent, and incongruent. During the task, participants will be presented with a central arrow target that can appear either above or below a cross-fixation point. The target arrow will be surrounded by two flankers on each side.

3. *Executive function*: To assess executive function, the Category Verbal Fluency Test (CVFT) will be employed in this study. During this test, participants will be instructed to

verbally generate as many words as possible within a 60-second timeframe for three specific categories: animals, fruits, and vegetables. The participants will overtly produce words within each category, aiming to generate as many correct responses as they can within the given time limit. The total number of correct words generated across the three categories will serve as a measure of executive function [22].

**Secondary outcomes.**

1. Objective assessment of circadian rhythms: Actigraphic records are used to quantify the sleep-wake cycle and estimate the objective sleep efficiency [34]. We use the Philips Actiwatch2, typically the size of a wristwatch, continuously for multiple days and nights during the interventions. This objective measurement of sleep quality provides key metrics such as Bedtime, Get Up Time, Time in Bed (hours), Total Sleep Time (hours), Onset Latency (minutes), Sleep Efficiency (percent), and Wake After Sleep Onset (WASO, minutes). These daily sleep statistics allow for the derivation of additional parameters to further evaluate circadian rhythms and sleep quality.

2. Global cognition will be evaluated using the Montreal Cognitive Assessment Hong Kong version (HK MoCA). This assessment tool is widely recognized for its validity in detecting early cognitive dysfunction in individuals with neurocognitive disorders [22].

## Treatment schedule

The treatment schedule consists of a 2-week course with 5 sessions per week, each session lasting 20 minutes. All participants, regardless of their assigned group, will undergo a total of 10 treatment sessions. Notably, the treatment schedules are identical for all three randomized groups in the study.

## Statistical analysis

To ensure unbiased analysis, the data analyst involved in this study will be blinded to the grouping of participants. The data analysis will follow an intention-to-treat principle. Linear mixed models will be employed to examine the variances between conditions concerning both primary and secondary outcome measures at each designated time point. This statistical approach enables the incorporation of participants with incomplete data. The fixed effects in the models will encompass treatment, time points, and their interaction, while participants will be considered random effects at various time points. Comparisons between the randomized groups will be conducted to assess pre-treatment, cognitive performance and the function of the glymphatic system. The analysis will include evaluating the changes in sleep quality and cognitive functions across the randomized groups from baseline to follow-up points. This analysis will be conducted by considering the time points as level one and participants as level two. Additionally, covariates identified from the differences observed at baseline will be included in the regression model. Secondary analyses will involve examining group differences in domain-specific function outcomes, as well as exploring the association between changes in Pittsburgh Sleep Quality Index (PSQI) scores and both cognitive functions and the function of the glymphatic system. The study will also monitor the occurrence of adverse events and assess program adherence characteristics. Statistical significance will be determined using a 2-sided p-value of less than 0.05. Computation and analysis will be performed using R Studio (version 1.1.456).

## Study timeline

The trial has been scheduled to take place from the 1st of January 2024 to the 24th of June 2025. By the end of February 2024, we have successfully screened thirty-four eligible participants of which 16 participants have completed comprehensive baseline assessments and structural MRI scanning.

## Discussion

This paper presents the rationale and study protocol for a pilot randomized sham-controlled trial aiming to assess the effects of hf-tRNS and 40 Hz tACS interventions on subjective sleep quality, sleep-awake cycles, and cognitive functions in individuals with mild vascular cognitive impairment (VCI). This study hypothesizes that these interventions will provide greater benefits in these areas. Additionally, it is expected that the improvements in sleep quality and cognitive functions will be associated with enhanced functioning of the glymphatic system, as measured by DTI. This trial adopts a neuroscience-driven approach to address the non-pharmacological management of a common comorbidity in elderly patients. Hf-tRNS and 40 Hz tACS interventions offer the advantage of modulating neuronal activities through frequency-specific currents, effectively stimulating the endogenous neural oscillations and regulating circadian rhythms. As far as we know, this study is the first randomized clinical trial to compare the safety, efficacy, and sustainability of hf-tRNS and 40 Hz tACS in the treatment of sleep disturbances in individuals with mild VCI. To ensure the highest methodological quality, the trial will adhere to the CONSORT checklist. Eligible participants will undergo randomization administered by an impartial assessor who is unaware of the distinctions among the three modalities of interventions (i.e., hf-tRNS, 40 Hz tACS, and sham tCS). Throughout the study period, the assessors and statisticians involved in the study will not interact with the participants and will continue to be blinded by group assignments. The results of this study will provide valuable clinical evidence regarding the application of oscillation-specific transcranial current stimulation for modulating glymphatic function and treating sleep disturbances in preclinical vascular dementia. Regardless of the outcome aligning with or deviating from the study hypothesis, the results will be shared through international peer-reviewed journals, seminars, and scientific conferences. Furthermore, all participants and their families will be informed about the study's findings.

## Conclusion

The outcomes of this study will yield valuable and robust evidence regarding the impact of oscillation-specific modalities of transcranial current stimulation on sleep, glymphatic function, and cognition. The findings will contribute to advancing our understanding of the interplay between these factors and their relevance to sleep medicine and neurovascular diseases. Ultimately, the results will have far-reaching implications for the development of innovative interventions and therapeutic approaches in sleep medicine and the management of neurovascular conditions.

## Supporting information

**S1 Checklist. SPIRIT 2013 checklist: Recommended items to address in a clinical trial protocol and related documents\*.**
(PDF)

**S1 Fig. The Consolidated Standards of Reporting Trials (CONSORT) flow diagram for the clinical trial of oscillation-specific transcranial current stimulation in mild vascular**

**cognitive impairment patients.**
(TIF)

**S2 Fig. Schedule of the clinical trial according to the standard protocol items: Recommendations for interventional trials checklist (SPIRIT).** Abbreviations: hf-tRNS = High-frequency transcranial random noise stimulation; tACS = Transcranial alternating current stimulation; tCS = Transcranial current stimulation.
(TIF)

**S1 File. Research protocol.**
(PDF)

## Author Contributions

**Conceptualization:** Hanna Lu.

**Funding acquisition:** Hanna Lu.

**Investigation:** Jing Li, Yuk Shan Yuen, Natalie Shu Yang.

**Methodology:** Natalie Shu Yang, Zeyan Li, Wai Kwong Tang, Hanna Lu.

**Writing – original draft:** Yuqi Gong.

**Writing – review & editing:** Yuqi Gong, Hanna Lu.

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
