## [Decision Letter · Decision Letter 0]

11 Jun 2024

PONE-D-24-08334Daily high-frequency transcranial random noise stimulation (hf-tRNS) for sleep disturbances and cognitive dysfunction in patients with mild vascular cognitive impairments: a study protocol for a pilot randomized controlled trialPLOS ONE

Dear Dr. Lu,

Thank you for submitting your manuscript to PLOS ONE. After careful consideration, we feel that it has merit but does not fully meet PLOS ONE’s publication criteria as it currently stands. Therefore, we invite you to submit a revised version of the manuscript that addresses the points raised during the review process.

**ACADEMIC EDITOR: **Please clearly specify the anticipated start date of this study in the main text of the manuscript, the specific registration date on ClinicalTrials.gov, and the current latest stage of the research.==============================

We look forward to receiving your revised manuscript.

Kind regards,

Hantong Hu, M.D

Academic Editor

PLOS ONE

2. For studies involving third-party data, we encourage authors to share any data specific to their analyses that they can legally distribute. PLOS recognizes, however, that authors may be using third-party data they do not have the rights to share. When third-party data cannot be publicly shared, authors must provide all information necessary for interested researchers to apply to gain access to the data. (https://journals.plos.org/plosone/s/data-availability#loc-acceptable-data-access-restrictions)

a) A description of the data set and the third-party source

b) If applicable, verification of permission to use the data set

c) Confirmation of whether the authors received any special privileges in accessing the data that other researchers would not have

d) All necessary contact information others would need to apply to gain access to the data

Reviewers' comments:

Reviewer's Responses to Questions

**Comments to the Author**

1. Does the manuscript provide a valid rationale for the proposed study, with clearly identified and justified research questions?

Reviewer #1: Yes

Reviewer #2: Yes

2. Is the protocol technically sound and planned in a manner that will lead to a meaningful outcome and allow testing the stated hypotheses?

Reviewer #1: Yes

Reviewer #2: Yes

3. Is the methodology feasible and described in sufficient detail to allow the work to be replicable?

Reviewer #1: Yes

Reviewer #2: Yes

4. Have the authors described where all data underlying the findings will be made available when the study is complete?

Reviewer #1: Yes

Reviewer #2: No

5. Is the manuscript presented in an intelligible fashion and written in standard English?

Reviewer #1: Yes

Reviewer #2: Yes

6. Review Comments to the Author

You may also provide optional suggestions and comments to authors that they might find helpful in planning their study.

Reviewer #1: 1. Please explain clearly what HF and LF describes and its relevance to the Autonomic Nervous System. In the 1960s Dr Paul Nogier of GLEM ( Lyon) described and demonstrated how oscillations and corrections were necessary for wellbeing whilst teaching the central mechanisms of auricular medicine.

2. One needs to elaborate further the correlations between the lack of night sleep ( insomnia etc) and chronic inflammation and the negative outcomes on health and wellbeing. It is not the lack of sleep that is the main issue- it is the resultant chronic inflammatory response which is the major concern to the brain and body.

3.The left parietal lobe is described as the important region for measuring the changes. Why only the left parietal lobe?

There is no comment on changes possible at the SCN - home of sleep regulation / Alzheimers D / Parkinsons D. Were these regions explored? Could there be a dysregulation of the functional connectivity between LPL and SCN which contributes to the sleep and cognitive changes?

Reviewer #2: Its an interesting study to look at different modalities for potentially boosting cognitive function. However, MOCA is a rather low sensitivity instrument to assess for change compared to a more comprehensive neuropsychological test battery.

Also, please provide details of stimulation parameters in the sham group and how you have tested to assess whether it produces enough local irritation to serve as an effective sham.

7. PLOS authors have the option to publish the peer review history of their article (what does this mean?). If published, this will include your full peer review and any attached files.

Reviewer #1: **Yes: **IM QUAH-SMITH

Reviewer #2: No

---

## [Author Response · Author response to Decision Letter 0]

5 Jul 2024

Reviewer #1: Dr. Im Quah-smith

Comment 1. 

Please explain clearly what HF and LF describes and its relevance to the Autonomic Nervous System. In the 1960s Dr Paul Nogier of GLEM (Lyon) described and demonstrated how oscillations and corrections were necessary for wellbeing whilst teaching the central mechanisms of auricular medicine.

Response 1. 

Thank you for your question regarding the description and relevance of HF (high-frequency) and LF (low-frequency) stimulation and their connection to the autonomic nervous system (ANS). We appreciate your interest in this topic and the opportunity to provide clarification.

HF or LF in our paper refer to specific frequency ranges used in transcranial random noise stimulation (tRNS). Conventionally, LF-tRNS is administered within the range of 0.1 Hz to 100 Hz, while HF-tRNS is delivered within the range of 101 Hz to 640 Hz (Terney, 2008). In our paper, the specific frequency range used for HF-tRNS is between 101 Hz and 640 Hz. We have updated the details of high-frequency stimulation on page 11 line 280-285.

Indeed, the effects of transcranial random noise stimulation (tRNS) on regulating the autonomic nervous system (ANS) are still relatively limited in research, and there is insufficient evidence to support its direct impact on the ANS. Most studies related to tRNS primarily focus on its effects on brain function and neuroplasticity. However, the pioneered studies of Dr. Paul Nogier may lead to another critical modality of neuromodulation, named transcutaneous vagus nerve stimulation (tVNS). The electrical impulses delivered by tVNS device could activate specific vagal pathways, triggering a cascade of physiological responses. One of the primary effects of tVNS is the modulation of neurotransmitters, such as norepinephrine, which play vital roles in the regulation of autonomic nervous system. Although we could not include tVNS in current pilot clinical trial, this topic is a definitely valuable idea that is worthy to explore in our next trial.

Reference

Terney D, Chaieb L, Moliadze V, Antal A, Paulus W. Increasing human brain excitability by transcranial high-frequency random noise stimulation. Journal of Neuroscience. 2008;28(52):14147–14155.

Comment 2. 

One needs to elaborate further the correlations between the lack of night sleep (insomnia etc) and chronic inflammation and the negative outcomes on health and wellbeing. It is not the lack of sleep that is the main issue- it is the resultant chronic inflammatory response which is the major concern to the brain and body.

Response 2. 

Thank you for your insightful comment regarding the interconnections between insomnia, chronic inflammation, and their adverse implications for health and well-being. We appreciate your emphasis on the significance of chronic inflammation as a substantial concern impacting the brain and overall bodily functions.

Based on your comment, we have done a brief literature review. Indeed, sleep plays a pivotal role in modulating cytokine levels and leading to inflammatory response. Sleep deprivation has been shown to activate inflammatory pathways and trigger the release of cytokines, such as interleukin-6 (IL-6) and tumor necrosis factor-alpha (TNF-alpha) (Besedovsky et al., 2012). Moreover, recent research has demonstrated that salivary biomarkers exhibit higher reliability, with increased intra-class correlation coefficient (ICC) for inflammatory markers compared to oxidative stress markers (Nam, 2019). However, the findings regarding IL-6, IL-1β, and TNF-alpha in saliva are diverse, with a study have reported elevated secretion of various inflammatory markers, including C-reactive protein, IL-6, and TNF-alpha, in relation to reduced or disrupted sleep (Rico-Rosillo, 2018). Some studies reporting an increase in IL-6 but a decrease in IL-1β and TNF-alpha (Zhang, 2020), while others found an increase in both TNF-alpha and IL-1β (Pinto, 2016). Some studies observed no significant association between TNF-alpha and IL-1β levels and sleep quality (Reinhardt, 2019), while another study reported an increase in IL-1β levels (LaVoy, 2020). Despite the intricacies surrounding these findings, chronic inflammatory responses have been linked to various health conditions, including cardiovascular disease, metabolic disorders, neurodegenerative diseases, mood disorders, and impaired cognitive function (Irwin, 2019; Prather et al., 2017). Given the complexity of these relationships, it is crucial to consider the role of chronic inflammatory in sleep disturbances and continue conducting research in this field to further comprehend the intricate association between sleep quality and salivary cytokines. 

Therefore, we have updated this part in the discussion and future directions highlighting the intricate correlations between sleep deprivation, chronic inflammation, and their profound impact on health and well-being on page 2, line 26 - 44. Meanwhile, for enhancing the assessment of night sleep, we employed the proxy of sleep efficiency for measuring individual’s night sleep quality and duration using the formula as: Duration of night sleep/Hours in bed * 100%.

References

Besedovsky L, Lange T, Born J. Sleep and immune function. Pflügers Archiv-European Journal of Physiology. 2012;463(1):121–137.

Nam Y, Kim YY, Chang JY, Kho HS. Salivary biomarkers of inflammation and oxidative stress in healthy adults. Archives of Oral Biology. 2019;97:215–222.

Rico-Rosillo MG, Vega-Robledo GB. Sleep and immune system. Revista Alergia Mexico. 2018;65(2):160–170.

Zhang L, Zhang R, Shen Y, Qiao S, Hui Z, Chen J. Shimian granules improve sleep, mood and performance of shift nurses in association changes in melatonin and cytokine biomarkers: a randomized, double-blind, placebo-controlled pilot study. Chronobiology international. 2020;37(4):592–605.

Pinto AR, Da Silva NC, Pinato L. Analyses of melatonin, cytokines, and sleep in chronic renal failure. Sleep and Breathing. 2016;20:339–344.

Reinhardt ´EL, Fernandes PACM, Markus RP, Fischer FM. Night work effects on salivary cytokines TNF, IL-1β and IL-6. Chronobiology international. 2019;36(1):11–26.

LaVoy EC, Palmer CA, So C, Alfano CA. Bidirectional relationships between sleep and biomarkers of stress and immunity in youth. International Journal of Psychophysiology. 2020;158:331–339.

Irwin MR. Sleep and inflammation: partners in sickness and in health. Nature Reviews Immunology. 2019;19(11):702–715.

Prather AA, Gurfein B, Moran P, Daubenmier J, Acree M, Bacchetti P, et al. Tired telomeres: poor global sleep quality, perceived stress, and telomere length in immune cell subsets in obese men and women. Brain, behavior, and immunity. 2015;47:155–162.

Comment 3. 

The left parietal lobe is described as the important region for measuring the changes. Why only the left parietal lobe? There is no comment on changes possible at the SCN - home of sleep regulation / Alzheimers D / Parkinsons D. Were these regions explored? Could there be a dysregulation of the functional connectivity between LPL and SCN which contributes to the sleep and cognitive changes?

Response 3. 

Thank you for your valuable comment on why we selected left parietal lobe and whether we explored other regions like the suprachiasmatic nucleus (SCN) in sleep regulation. 

As you highlighted, we acknowledge the crucial role of the SCN in circadian rhythms and sleep regulation and explain the rationale of selecting left parietal lobe as treatment target on page 10. The SCN, as the master circadian clock, interacts with various brain regions to regulate sleep-wake cycles. Emerging evidence suggests that the SCN and default mode network (DMN) show synchrony in regulating the arousal state of the brain system (Kaufmann, 2006). Furthermore, abnormal functional connectivity between SCN and DMN may be an underlying pathophysiological mechanism of sleep-wake disorders (Kyeong et al., 2017). Since SCN is located in the deep structures of the brain, transcranial current stimulation is unable to deliver the stimulation to deep structures. Focusing on the DMN, the brain regions involved in DMN include: medial prefrontal cortex, posterior cingulate cortex and inferior parietal lobule (IPL). Among the three brain regions, prefrontal cortex and posterior cingulate cortex are located at the medial surface of brain, which is unable to stimulate through weak current stimulation. IPL is located on the outer surface of the brain, which makes it a good candidate for the stimulation target. Besides, previous studies have showed that the LPL is involved in various cognitive processes that are often disrupted in neurodegenerative diseases, making it a focal point for our investigation of cognitive changes (Buckner et al., 2008). Thus, we updated this part in the rationale of selecting IPL as treatment target on Page 10 Line 260 - 272. We appreciate your insightful comment and recognize the importance of exploring the functional connectivity between the LPL and the SCN. We will consider incorporating these aspects into our future studies to gain a more comprehensive understanding of the complex relationship between brain regions involved in sleep regulation, cognitive changes, and neurodegenerative diseases. Addressing these connections could provide valuable insights into potential therapeutic targets for improving cognitive and sleep-related outcomes in affected patients.

Reference

Kaufmann C, Wehrle R, Wetter T, Holsboer F, Auer D, Pollmächer T, et al. Brain activation and hypothalamic functional connectivity during human non-rapid eye movement sleep: an EEG/fMRI study. Brain. 2006;129(3):655–667.

Kyeong S, Choi SH, Shin JE, Lee WS, Yang KH, Chung TS, et al. Functional connectivity of the circadian clock and neural substrates of sleep-wake disturbance in delirium. Psychiatry Research: Neuroimaging. 2017;264:10–12.

Buckner RL, Andrews-Hanna JR, Schacter DL. The brain’s default network: anatomy, function, and relevance to disease. Annals of the New York Academy of Sciences. 2008;1124(1):1–38.

Reviewer #2: 

Comment 1. 

It’s an interesting study to look at different modalities for potentially boosting cognitive function. However, MOCA is a rather low sensitivity instrument to assess for change compared to a more comprehensive neuropsychological test battery. 

Response 1. 

Thank you for your feedback on our study. We appreciate your interest and valuable input regarding the assessment of cognitive function. We have carefully considered your comments and would like to address your concerns.

We understand that you have raised a point about the sensitivity of the Montreal Cognitive Assessment (MoCA) in assessing change compared to a more comprehensive neuropsychological test battery. We agree that a broader range of assessments can provide a more thorough evaluation of cognitive function. In our study, we have incorporated multiple cognitive tests to complement the MoCA and enhance the validity of our findings on page 12 line 310 - 336.

To assess subjective sleep quality, we utilized the Pittsburgh Sleep Quality Index (PSQI), a self-report questionnaire consisting of 19 items. The PSQI provides a measure of global subjective sleep quality, and we employed the validated Chinese version of the PSQI. For attentional function, we employed the Attention Network Test (ANT), which measures complex attention. The ANT paradigm, implemented using E-Prime 3.0 software, includes different cues and flankers to evaluate attentional networks. To assess executive function, we utilized the Category Verbal Fluency Test (CVFT). Participants were instructed to generate as many words as possible within a 60-second timeframe for specific categories, such as animals, fruits, and vegetables. The total number of correct words generated across the three categories served as a measure of executive function.

In addition to these cognitive assessments, we included objective measures to evaluate circadian rhythms and global cognition. Actigraphic records were used to quantify the sleep-wake cycle and estimate objective sleep efficiency. This objective measurement provides information on sleep quality and the sleep-wake cycle, allowing for the assessment of circadian rhythms in the participants.

Comment 2. 

Also, please provide details of stimulation parameters in the sham group and how you have tested to assess whether it produces enough local irritation to serve as an effective sham.

Response 2. 

Thank you for your question regarding the stimulation parameters in the sham group and the assessment of local irritation to ensure its effectiveness as a sham condition. We appreciate the opportunity to provide additional details on this aspect of our study.

In the sham group, the stimulation modality employed was sham transcranial current stimulation (tCS). The duration of the sham stimulation was limited to 30 seconds, and the electrode placement was identical to that of the active stimulation groups. This ensured that the participants in the sham group experienced the same electrode positioning and initial sensations as the active stimulation groups.

To assess the effectiveness of the sham stimulation in producing enough local irritation, we aimed to replicate the temporary skin sensation by keeping the electrodes in place for an additional 20 minutes following the 30-second sham stimulation. So that we can ensure that participants in the sham group experienced a similar tingling sensation as the active stimulation groups. This approach allowed us to maintain blinding and provide an effective control condition for comparison with the active stimulation groups.

We acknowledge that providing a more specific assessment or measure to evaluate local irritation would have strengthened our study. In future study, we will consider incorporating additional information on any specific assessments conducted to evaluate the effectiveness of the sham stimulation in producing local irritation.

---

## [Decision Letter · Decision Letter 1]

24 Jul 2024

PONE-D-24-08334R1Daily high-frequency transcranial random noise stimulation (hf-tRNS) for sleep disturbances and cognitive dysfunction in patients with mild vascular cognitive impairments: a study protocol for a pilot randomized controlled trialPLOS ONE

Dear Dr. Lu,

Thank you for submitting your manuscript to PLOS ONE. After careful consideration, we feel that it has merit but does not fully meet PLOS ONE’s publication criteria as it currently stands. Therefore, we invite you to submit a revised version of the manuscript that addresses the points raised during the review process.

We look forward to receiving your revised manuscript.

Kind regards,

Hantong Hu

Academic Editor

PLOS ONE

Journal Requirements:

Reviewers' comments:

Reviewer's Responses to Questions

**Comments to the Author**

1. Does the manuscript provide a valid rationale for the proposed study, with clearly identified and justified research questions?

Reviewer #2: Yes

2. Is the protocol technically sound and planned in a manner that will lead to a meaningful outcome and allow testing the stated hypotheses?

Reviewer #2: Yes

3. Is the methodology feasible and described in sufficient detail to allow the work to be replicable?

Reviewer #2: Yes

4. Have the authors described where all data underlying the findings will be made available when the study is complete?

Reviewer #2: Yes

5. Is the manuscript presented in an intelligible fashion and written in standard English?

Reviewer #2: Yes

6. Review Comments to the Author

You may also provide optional suggestions and comments to authors that they might find helpful in planning their study.

**Reviewer #2:** Please clarify "skin physiological test" under sham method. For sham group, consider having a low amplitude stimulation for longer in sham group as the skin sensation is caused by current as well as the application of electrodes.

Have you considered the inaccuracy of actigraphy in assessing sleep and how you will address in data interpretation. You can consider including sleep diary. Sleep diaries are also not accurate, but having redundancy in sleep assessment could be helpful. It will also provide information about subjective improvement or worsening from the patient's perspective.

7. PLOS authors have the option to publish the peer review history of their article (what does this mean?). If published, this will include your full peer review and any attached files.

Reviewer #2: No

---

## [Author Response · Author response to Decision Letter 1]

5 Aug 2024

Thank you for your thoughtful and constructive comments on our manuscript. We appreciate your insights and the opportunity to clarify and improve our study.

Regarding the "skin physiological test" mentioned in the sham method, we apologize for the lack of clarity. This test is a preliminary assessment designed to ensure that the application of electrodes and any potential low-level electrical stimulation do not induce unintended physiological changes in the skin. Specifically, we monitor skin temperature and any signs of irritation or discomfort at the electrode sites before and after the sham stimulation. The primary goal is to confirm the integrity of the skin and ensure that the sham method does not produce any physiological effects that could confound the study results.

We also appreciate your suggestion to consider low amplitude stimulation for the sham group. Incorporating a low amplitude stimulation for a longer duration can better mimic the sensation of active stimulation without delivering therapeutic current levels. This modification will enhance the effectiveness of participant blinding and ensure the integrity of the sham condition. Given that we have completed the training of experimental setting, we will definitely implement this sham condition in our future clinical trials to maintain consistency.

Additionally, we acknowledge the limitations of actigraphy in accurately assessing sleep patterns and appreciate your suggestion to address this issue. Currently, we use the Philips Actiwatch2, which provides key metrics such as Bedtime, Get Up Time, Time in Bed (hours), Total Sleep Time (hours), Onset Latency (minutes), Sleep Efficiency (percent), and Wake After Sleep Onset (WASO, minutes). These daily sleep statistics allow for the derivation of additional parameters to further evaluate circadian rhythms and sleep quality. The above contents have been updated on Page 13, Paragraph 2.

The Philips Actiwatch monitoring system has been validated for reliable sleep record-keeping (Sadeh, 2011; Roomkham, 2019; Menghini, 2024). Although the PSQI scale can cover the period participants wore the Actiwatch, we recognize the value of including a sleep diary to complement actigraphy data. Combining actigraphy with self-reported sleep data will provide a more comprehensive view of the participants' sleep patterns. This approach will help mitigate the inaccuracies inherent in each method and improve the reliability of our sleep assessment. We agree that using a sleep diary alongside actigraphy in this study can enhance the assessment of sleep quality. However, if we add sleep diary into treatment outcomes, then there will be too many outcome measures in a pilot study. Of course, we will use sleep diary in the full-size clinical trial. This will allow us to capture subjective improvements or worsening in sleep from the patients' perspective, which is essential for understanding the broader impact of our intervention.

We believe these modifications will address your concerns and enhance the rigor and reliability of our study. Thank you once again for your valuable feedback

References:

Roomkham S, Hittle M, Cheung J, Lovell D, Mignot E, Perrin D. Sleep monitoring with the Apple Watch: Comparison to a clinically validated actigraph. F1000Research. 2019;8:754.

Menghini L, Balducci C, de Zambotti M. Is it Time to Include Wearable Sleep Trackers in the Applied Psychologists’ Toolbox? The Spanish Journal of Psychology. 2024;27:e8.

Sadeh A. The role and validity of actigraphy in sleep medicine: an update. Sleep Medicine Reviews. 2011;15(4):259-267.

---

## [Editor Report · Decision Letter 2]

8 Aug 2024

Daily high-frequency transcranial random noise stimulation (hf-tRNS) for sleep disturbances and cognitive dysfunction in patients with mild vascular cognitive impairments: a study protocol for a pilot randomized controlled trial

PONE-D-24-08334R2

Dear Dr. Lu,

We’re pleased to inform you that your manuscript has been judged scientifically suitable for publication and will be formally accepted for publication once it meets all outstanding technical requirements.

Kind regards,

Hantong Hu

Academic Editor

PLOS ONE

Additional Editor Comments (optional):

Thank you to the authors for the careful revisions during the two rounds. I believe this manuscript now meets the publication standards and recommend it for publication.
---

## [Editor Report · Acceptance letter]

15 Aug 2024

PONE-D-24-08334R2 

PLOS ONE

Dear Dr. Lu, 

I'm pleased to inform you that your manuscript has been deemed suitable for publication in PLOS ONE. Congratulations! Your manuscript is now being handed over to our production team.

Kind regards, 

on behalf of

Dr. Hantong Hu 

Academic Editor

PLOS ONE